# Structural Insights into Neonicotinoids and N-Unsubstituted Metabolites on Human nAChRs by Molecular Docking, Dynamics Simulations, and Calcium Imaging

**DOI:** 10.3390/ijms241713170

**Published:** 2023-08-24

**Authors:** Karin Grillberger, Eike Cöllen, Claudia Immacolata Trivisani, Jonathan Blum, Marcel Leist, Gerhard F. Ecker

**Affiliations:** 1Department of Pharmaceutical Sciences, University of Vienna, 1090 Vienna, Austria; 2In Vitro Toxicology and Biomedicine, University of Konstanz, 78457 Konstanz, Germany; 3Department of Biotechnology, Chemistry and Pharmacy, University of Siena, 53100 Siena, Italy

**Keywords:** neonicotinoids, pesticides, metabolites, nAChR, docking, molecular dynamics simulations, calcium imaging

## Abstract

Neonicotinoid pesticides were initially designed in order to achieve species selectivity on insect nicotinic acetylcholine receptors (nAChRs). However, concerns arose when agonistic effects were also detected in human cells expressing nAChRs. In the context of next-generation risk assessments (NGRAs), new approach methods (NAMs) should replace animal testing where appropriate. Herein, we present a combination of in silico and in vitro methodologies that are used to investigate the potentially toxic effects of neonicotinoids and nicotinoid metabolites on human neurons. First, an ensemble docking study was conducted on the nAChR isoforms α7 and α3β4 to assess potential crucial molecular initiating event (MIE) interactions. Representative docking poses were further refined using molecular dynamics (MD) simulations and binding energy calculations using implicit solvent models. Finally, calcium imaging on LUHMES neurons confirmed a key event (KE) downstream of the MIE. This method was also used to confirm the predicted agonistic effect of the metabolite descyano-thiacloprid (DCNT).

## 1. Introduction

Complex toxicological endpoints and their adequate assessment are major challenges in the field of predictive toxicology. Moreover, this field of research is currently facing a paradigm shift, leading towards alternatives to classical animal testing strategies. Next-generation risk assessments (NGRAs) are a new way to assess the assurance of the safety of chemicals and drugs by applying modern, human-centered approaches. This implies the use of new approach methodologies (NAMs), such as in silico and in vitro tools, where appropriate. In this context, the implementation of adverse outcome pathways (AOPs) is very useful, since they link a molecular initiating event (MIE) to several key events (KEs), which eventually leads to adverse toxicological outcomes. Modern tools, such as QSAR, machine learning, and artificial intelligence, play an increasing role in (computational) toxicology and, thus, in NGRAs [1,2,3].

Evidence from previous studies suggests that exposure to environmental toxicants can lead to developmental neurotoxicity (DNT), which can manifest in neurodevelopmental disorders like autism, an increased incidence of attention deficit/hyperactivity disorder (ADHD), or cognitive deficits [4,5]. Addressing the urgent need for an integrated approach to testing and assessment (IATA) for DNT [6], an in vitro test battery was established recently [7]. There is already an AOP framework naming eight MIEs that are linked to DNT [8]. However, the list of target proteins involved in different cell signaling pathways is by no means comprehensive [9]. Some of the proteins involved in the MIE of DNT comprise neurotransmitter receptors and ion channels. For instance, nicotinic acetylcholine receptors (nAChRs) could be putative targets. Furthermore, the underlying gene was mentioned in a general list for developmental and reproductive toxicology (DART) [10]. These ligand-gated ion channels are members of the cys-loop-receptor family, including the 5-HT3, glycine, and GABA receptors, and they are, therefore, composed of five homo- or heteromeric subunits. The central pore becomes permeable for ions upon agonist binding to the orthosteric binding site, which is located between two adjacent subunits of the complex. A group of aromatic tyrosine and tryptophan residues, originating from loops A–C and D–F in the principal and complementary subunits, respectively, line the binding pocket. Notably, different conformational states are exhibited, depending on the activation state and/or the bound ligand [11].

Amongst the potentially triggering molecules for human nAChRs are a group of pesticides called neonicotinoids. Despite recent concerns regarding potential human exposure [12,13,14] and adverse effects [15,16], they are still widely used for urban and agricultural purposes. Originally, these chemicals were designed in order to achieve selective toxicity in insect receptors [15], but recent studies suggest that some of them probably also effect neuronal signaling in human cells [17,18]. In particular, nicotinoid metabolites, such as the desnitro derivative of imidacloprid, are suggested to display similar agonistic activities to the prototype neurotoxicant nicotine [17]. In this context, two IATA case studies (numbers 4 and 5), which were published in 2021, showcased novel methodologies that were used for the evaluations of acetamiprid, imidacloprid, and its desnitro metabolite [19]. Generally, regulatory authorities emphasize the importance of including metabolites and metabolism in toxicological risk assessments. This is particularly crucial considering the fact that there are some cases in which metabolism leads to toxification [20,21,22]. It is assumed that other neonicotinoids may also result in more active metabolites, like descyano-thiacloprid (DCNT) and descyano-thiacloprid-olefin (DCNTO). However, there are few published data available concerning these metabolized compounds in humans. Therefore, the aim of this study is to provide an increased understanding of whether and how further nicotinoid metabolites, DCNT and DCNTO, are also involved in one of the MIEs of DNT, which would be represented by binding to some of the most abundant isoforms of nAChRs, namely, α3β4 and α7. This relates to the DNT-associated AOPs 12 and 13 [23,24], since the underlying pathway starts with receptor binding, which leads to a chain of effects similar to the hypothesis proposed herein. The tiered approach presented herein combines in vitro and in silico tools. Molecular docking, molecular dynamics (MD) simulations, and binding energy calculations serve as computational tools that help to gain detailed insights into binding to human target proteins. Moreover, a suitable in vitro assay (calcium imaging on LUHMES cells) demonstrates that a triggering effect can be detected for the descyano metabolite DCNT at a cellular level, which is similar to the prototype agonist nicotine. This study aims to contribute to this growing area of research by using NAMs in the context of hazard identification and risk assessment, by exploring the strengths and limitations of combining in silico and vitro approaches.

## 2. Results and Discussion

### 2.1. Ensemble Docking Analysis

Molecular docking can serve as a structure-based tool to elucidate binding modes and to prioritize ligands that are more likely to bind to the protein of interest [25]. The docking algorithm places multiple conformations of the ligand into the previously defined binding site grid by exhaustively searching for possible locations and orientations. After refining the initially large number of docking poses, empirical scoring functions are applied. They account for favorable receptor–ligand interactions, penalize steric clashes or violations of physical principles, and are reported in kcal/mol [26,27]. Therefore, more negative values of docking scores indicate energetically more stable binding complexes. Depending on different (re-)scoring methods, alternative docking poses can be amongst the top-ranked ones that represent the best-docked version.

Within this study, 15 compounds (Appendix A) were docked to multiple conformational ensembles of the nAChR isoforms α3β4 and α7. The representative final docking modes in each protein structure were selected within the top 5 ranked poses based on the docking score and visual inspection to further investigate the binding modes. By assessing the docking performance of the ensemble structures using a Spearman correlation matrix, the ability to reproduce the experimental ranking is evaluated.

Docking to the α3β4 ensembles results in comparable Spearman rank correlations using the docking score and experimental IC_50_ [28,29] as metrics (Appendix A). Interestingly, the antagonist-bound structure including the co-crystallized water molecule in the binding site (indicated as water set 1, i.e., 6pv8-ws1) had the best performance, reporting a positive ranking correlation of 0.81 when the docking score was used as a metric (Table 1). 

Regarding the α7 isoform, three different conformational states of nAChR were available. The activated (PDB-ID: 7kox) and desensitized (PDB-ID: 7koq) structures are relatively similar with respect to the RMSD of the ECD (0.777, calculated using the align tool in Pymol [30]). The resting state of the receptor is represented by PDB-ID: 7koo. This structure is characterized by an open conformation of loop C, which leads to an increased size of the binding pocket. The result from ensemble docking indicates that the resting conformation of nAChR α7 has the worst ability to emulate experimentally derived binding affinities. This is evident from the low Spearman ranking correlation of docking score and delta G binding energy to experimentally derived IC_50_ values. A likely explanation for this lower correlation is that relatively small ligands are accommodated better in the binding pocket of the activated or desensitized state. In these conformations, the ligands exhibit a smaller solvent-accessible surface area (which is also considered in scoring functions) and have increased possibilities to form interactions with residues within the binding site. Nevertheless, it is important to also consider receptor flexibility, which can essentially not be examined when a semi-flexible docking protocol is applied. Moreover, studies on homologous acetylcholine binding proteins (AChBPs) lead to the assumption that, depending on the affinity of nAChR ligands, different conformational states are stabilized [31]. This implies that antagonists would rather stabilize the resting (like α-bungarotoxin in 7koo (α7)) or desensitized (7koq (α7), 6pv8 (α3β4)) state. On the contrary, agonists including nicotine and endogenous acetylcholine display a higher affinity for the activated (7kox (α7), 6pv7 (α3β4)) and desensitized state. Since the Spearman correlation of the resting state reported basically no ranking correlation (0.0089) between experimental IC_50_ and docking score, a putative hypothesis would be that active neonicotinoids show higher affinity for the activated and desensitized conformations of nAChR (Table 1). Therefore, due to the low Spearman correlation of PDB-ID 7koo, this structure was excluded from further refinement using MD simulations, since this strategy aims for in-depth characterization of the binding modes of small ligands. Another rationale for this decision was the better comparability between the two structures per nAChR isoform α7 (7kox, 7koq) and α3β4 (6pv7, 6pv8).

### 2.2. Ensemble Docking and Representative Binding Poses in nAChRs α7 and α3β4

#### 2.2.1. Imidacloprid (IMI)

Docking IMI to the structures of nAChR α7 and α3β4 revealed two distinct binding modes that are inverted to each other. One binding orientation is characterized by a similar orientation of the chloro-pyridine ring, as also reported by other nicotinoid compounds. This common binding mode was also co-crystallized (PDB-ID: 2zju [32]) with analogous AChBPs of the model organism *Lymnaea stagnalis* (great pond snail). In this binding mode, the heteroarylic ring is able to undergo a favorable π–π interaction with Trp148 (Trp149 in α3β4), which stabilizes the pyridine ring of nicotine. Additionally, halogen bonds would be possible in residues of the complementary subunit, namely Leu108, Gln116, and Leu118 in the α7 isoform. In α3β4, a halogen bond is only detected with the hydrophobic Leu123. Regarding the electronegative nitro moiety which points towards the tip of loop C, stabilizing cation–π interactions to the side chains of Trp148 and Tyr187 are predicted from the docking study of the α7 subtype. When this complex was subjected to a 50ns MD simulation, a hydrogen bond analysis revealed that an interaction with the sidechain of Gln56 of loop D would be possible in approximately 26% of the trajectory frames. In the heteromeric α3β4 isoform, the nitro group is reported to interact with two tyrosine residues from loop C at positions 190 and 197. Hydrogen bond analysis of the respective 50 ns trajectory identified interactions with Trp59, Trp149, and Tyr190 from the α3-chain and with Arg83 from the β4-chain in 7.78%, 16.37%, 5.39%, and 10.58% of the frames, respectively (Appendix A). 

Interestingly, a second binding mode was observed in the docking analysis that is essentially inverted to the above-described common mode. This inverted binding mode has previously been reported by photoaffinity labelling experiments, where the pyridine ring carried a photoactive group and contacted an amino acid in *Lymnaea stagnalis* [33], which is equivalent to Tyr167 in human nAChR α7. The docking analysis also suggested a second, inverted binding mode, where the pyridine ring is stabilized by three π–π stacking interactions to amino acids Trp54 of loop D, Trp148 of loop B, and Tyr92 of loop A. A hydrogen bond analysis of the α7-complex after simulating for 50 ns also showed that the backbone of Leu118 from the complementary subunit interacts in approximately 68% of the collected trajectories. Concerning the inverted mode of IMI in α3β4, Trp59 and Tyr197 are predicted to stabilize the pyridine ring via π–π interactions. Analysis of hydrogen bonds from the subjected trajectory reveals interactions with Tyr197 and Trp149 in 13.77% and 14.37% of the frames, respectively (Appendix A). 

#### 2.2.2. Desnitro-Imidacloprid (DNIMI)

DNIMI exclusively showed common nicotine-like binding modes that were characterized by a positioning of the pyridine ring analogous to the one of co-crystallized epibatidine. In a similar way to the common binding mode of the parent compound IMI in α7, the chlorine substituent is presumed to undergo halogen bonding to Leu108 and Gln116. In α3β4, only Leu123 could serve as a halogen bonding partner. Regarding the imidazolidine ring, the protonated nitrogen is predicted to form cation–π interactions with side chains of Trp54, Trp148, and Tyr194, which is equivalently reported in α3β4 (Trp59, Trp149, and Tyr197). Moreover, a hydrogen bond is formed with Tyr92 in α7, which could also be confirmed by a MD trajectory analysis in 74.05% of the frames for tautomer-3 and in 99.85% for tautomer-2. Analysis from docking to α3β4 revealed also hydrogen bonds with Trp149 and Ser148, which could be maintained for 60.68% and 22.16% of the simulation time, respectively (Appendix A).

#### 2.2.3. Thiacloprid (THIAC)

THIAC also exhibited two distinct binding modes that are inverse to each other (Figure 1). The stacking partner for π–π interactions in the common binding mode is predicted to be Trp148 in the α7-isoform and is hence similar to what has been reported for nicotine and related structures. In α3β4, no stacking interaction was present in the common binding orientation, but an additional hydrogen bond with Tyr93 is observed. However, in the inverted binding mode, the respective stacking partner in α7 and α3β4 changes to Tyr194 and Tyr197 of loop C, respectively. Notably, Leu118 from the complementary subunit has been reported to form a hydrogen bond in this binding orientation for 5.1% of the simulated α7-trajectory (Appendix A). An equivalent interaction was not seen in the α3β4-isoform (Appendix A).

#### 2.2.4. Descyano-Thiacloprid (DCNT) and Descyano-Thiacloprid-Olefin (DCNTO)

In an analogous way to DNIMI, the chlorine substituent of DCNT is expected to form halogen bonds with Leu108 (also DCNTO) and Gln116 in the homomeric α7 subtype (Appendix A). The protonated nitrogen in the thiazolidine ring is predicted to interact with Trp148 and Tyr194 via cation–π interactions for both descyano-metabolites DCNT and DCNTO. Additionally, DCNTO could form a cation–π contact with Tyr187. Concerning DCNTO, the side chain of Trp148 could also serve as a stacking partner for a π–π interaction with the thiazole ring. In the DCNT-α7 complex, hydrogen bonds are possible with Tyr92 and Trp148, which are stable for 49.90%/34.33% and 63.87%/51.30% of the MD simulation time (7koq/7kox). DCNTO forms hydrogen bonds with the backbone of Trp148 for 70.66% and of Tyr92 for 34.33%/5.39% of the simulation time (7koq/7kox). From docking to the α3β4 isoform, DCNTO shows π–π interactions with Trp59 and Trp149 from the principal subunit (Figure 2). Cation–π interactions are predicted for both DCNTO and DCNT with aromatic residues Trp59, Trp149, and Tyr197. Analyzing the respective trajectory for hydrogen bonds confirmed interactions with Trp149 for 54.49/63.67% of the frames in the case of DCNTO and for 55.09/73.05% in the DCNT complex (6pv7/6pv8).

To summarize, nicotinoid metabolites DNIMI, DCNT, and DCNTO are predicted to have more cation–π interactions with the residues of the aromatic cage, which is similar to the prototype developmental neurotoxicant nicotine [34]. Additionally, the thiazolyl ring of DCNTO provides properties that would enable π–π stacking interactions with these amino acids. Subsequent hydrogen bond analysis from the MD trajectories showed a similar pattern for both nAChR isoforms α7 and α3β4; contacts with Trp148, Tyr92, Trp149, and Tyr93 are predicted to be stable for a minimum of 50% of the simulation time. Contrary to this, the parent compounds IMI and THIAC showed less consistent contacts with these residues. However, the inverted binding mode from these two neonicotinoids in α7 exhibited contacts between their electronegative nitro or cyano group and the nonpolar Leu118.

### 2.3. Molecular Dynamics (MD) Simulation Analysis

In order to investigate the stability of the docked complexes of neonicotinoids (IMI and THIAC) and nicotinoid metabolites (DNIMI, DCNT, and DCNTO), the representative docking poses were further refined using MD simulations. Moreover, we aimed to investigate the proposed subtype selectivity for α3β4 over α7. The benefit of MD is that it simulates physiological conditions, including solvent and ions. This helps to assess protein–ligand complex stability and to confirm the binding hypotheses and main interactions. Finally, binding free energy calculations using an implicit solvent model are conducted as indicators for binding affinities [35,36].

#### RMSD Calculations

Analysis of RMSD (root mean square deviation) as a distance metric for the cartesian coordinates of structural models can be performed for different sets of reference selections. In the context of stability assessments of protein–ligand complexes, the mean RMSD of the molecule in the simulated trajectory of the complex can be considered as an insightful metric. 

From the mean RMSD analysis (Table 2), it is evident that the nicotinoid metabolites (DNIMI, DCNT, and DCNTO) have an overall lower mean RMSD in comparison to the parent compounds (IMI and TIHAC). This pattern applies for both nAChR models; in α7 ranging from 1.29 to 2.34 versus 2.14 to 3.87 and in α3β4 ranging from 1.37 to 2.46 versus 2.53 to 4.82, respectively. Moreover, the mean RMSDs of the inverted binding modes of the neonicotinoids have lower values in both nAChR isoforms, indicating that this orientation tends to be more stable in the binding site when compared to the common binding mode. Additionally, RMSD plots of the two different starting conformations per nAChR isoform are shown in Appendix A. 

Generally, pairwise RMSD calculations provide additional information about related conformational states which are sampled during MD simulations. This is performed by calculating the RMSD as a distance metric of the molecule to each frame of the same or different trajectory. In this way, 2D (pairwise) RMSD plots add information regarding conformational convergence. Off-diagonal peaks of higher RMSDs (green) indicate that previous states are not re-visited during the simulation. From the pairwise RMSD of the trajectory of IMI in nAChR α7 (7kox-ws1), which exhibits a common starting orientation, it is evident that two major conformational states are sampled in the 50 ns simulation time, with one major transition occurring at around 300 frames (after 30 ns) (Figure 3A,C). 

When the pairwise RMSD of two different trajectories is calculated, the similarity of two conformational ensembles can be evaluated. These plots do not necessarily result in a zero along the diagonal; however, blocks of low RMSD (blue) indicate that similar conformational states are sampled. This method is applied in order to assess whether the common and inverted binding modes are converging during the MD simulation. In the case of α7, two blocks of low RMSD are evident. The blue spot in the top left corner of Figure 3B would imply that, for the first 280 frames (28ns simulation time of the production run), the common mode is in a similar conformational space to the docked starting position of the inverted mode. Hence, the second blue block can be interpreted by the inverted mode of IMI being, for most of the simulation time, in a similar conformational space to the last 310 frames of the trajectory of the respective common binding mode. Therefore, two major conformational states are sampled, while the common mode is converging towards the inverted one at end of the simulated time.

### 2.4. nAChR Subtype Selectivity

Considering the fact that radioligand binding assays [29], as well as oocyte recordings of the nAChR subtypes α3β4 and α7 in *Xenopus laevis* [17], suggested increased subtype selectivity for the heteromeric isoform, we were interested in observing a similar effect in MD simulations. However, there was no clear pattern detected when using the post-processed binding energies as a metric. In the case of DNIMI, an average delta G (of the two different starting conformations) of −40.84 versus −36.31 kcal/mol was reported, which is in agreement with the suspected α3β4-subtype selectivity, but this was not the case for all six ligands.

When the binding sites of these two isoforms are aligned, a major difference arises from the change from polar Gln116 in α7 to nonpolar Leu121 in α3β4. When the structural surface of the agonist-bound structures is superimposed, Gln116 seems to lead to a more occluded binding site. Analysis of the MD trajectory of the inverted binding mode of THIAC and IMI highlighted an interaction with this residue, which might hint subtype selectivity. A previous study suggested an explanation for the inter-subtype selectivity of an antagonist towards α3β4 over α4β2 of nAChRs by a less compact binding site leading to better accommodation of the larger ligand, but also by weakening of hydrophobic van der Waals interactions of agonistic nicotine-like structures with aromatic box residues [37]. Regarding the suspected inferior affinity of homomeric α7, a reason might be the absence of an inter-subunit hydrogen bond between loops B and C, which was previously proposed by Grutter et al. in 2003 [38]. Additionally, nAChR α7 is known for rapid desensitization [11], which could also be a reason for the subtype selectivity which is observed in in vitro assays [28,29]. Combining the structural information, i.e., the absence of an inter-subunit hydrogen bond, with the fact that homodimeric receptors are known for rapid desensitization [11] might point towards the underlying mechanism for subtype selectivity.

### 2.5. MM-GBSA Binding Free Energy Calculations

#### 2.5.1. Postprocessing of Ensemble Docking Approach

Molecular-mechanics-based endpoint binding free energy calculations that apply an implicit solvent model can add additional value to a docking study. This approach can rescore generated docking poses and thus help to prioritize ligands with an increased binding affinity. As it is an empirical approach that is highly dependent on the force field that is used, cautious evaluation of the results is required. 

Schrödinger software has regularly updated available force fields, so in order to investigate substantial differences, the oldest (OPLS_2005 [39]) and the latest (OPLS4 [40]) were applied individually. The benefit of the former is a better comparability with previously published screening results, since it has been the standard method for many years, whereas the latter has improved treatment for molecular ions (i.e., a protonated nitrogen like in nicotine) as well as for sulfur interactions. Since the main ligands of interest (THIAC, DCNT, and DCNTO) all possess a sulfuric substructure, we assumed that the choice of the force field might have a substantial influence on the rescoring results and in consequence also on the understanding of the binding mechanisms.

When comparing the ranking correlations from the ensemble docking approach of the 15 ligands based on the delta G values, an inferior accuracy compared to the docking score was observed. Notably, here, the binding pose selection per ligand was based on the best glide_emodel score per ligand. By also incorporating a visual inspection and selecting from the top five scored poses, an improved ranking accuracy according to the delta G binding energy was achieved. Applying the OPLS4 and OPLS_2005 force field lead to Spearman correlations with experimental values of 1 and 0.9, respectively. Therefore, in the α7-isoform, the best Spearman correlation to the pIC_50_ values from a radioligand binding assay [28,29] was achieved when the OPLS4 force field was applied. This means ranking the (neo-)nicotinoid ligands from highest to lowest affinity (most negative to most positive dG binding energy) results in DCNTO > DCNT > DNIMI > THIAC > IMI. Interestingly, a very similar ranking order is predicted for the α3β4-subtype, where DCNT was slightly better scored than its olefin derivative.

From postprocessing using the MM-GBSA method [41] implemented in Schrödinger Software Suite version 21-1, it is also evident that the nicotinoid metabolites are predicted to have an overall higher binding affinity in terms of the delta G binding energy and docking score (Appendix A). This pattern is in agreement with the hypothesis that the loss of the nitro or cyano group of neonicotinoids through metabolism causes increased activity in human cells.

#### 2.5.2. Postprocessing of MD Simulations

Subjecting trajectories of MD simulations to MM-GB(PB)SA to analyze different energetic contributions of the ligand upon binding to the protein is a common practice in the field of computational drug design [35].

In this study, endpoint binding free energy calculations using the molecular mechanics generalized Born surface area (MM-GBSA) approach had better concordance with experimentally derived values than the Poisson–Boltzmann surface area (MM-PBSA) method. This is contrary to the results of predictions of the effects of mutations on ligand binding, where MM-PBSA had a better accuracy [42]. However, prediction of mutational effects has a greater focus on the properties of the protein, whereas the direction of research herein is concentrated more on the distinct binding modes of ligands. Nevertheless, it is important to assess both methods for a specific study in order to find the method that suits best.

We were able to observe a consistent pattern of nicotinoids (DNIMI, DCNT, and DCNTO) and prototype agonist nicotine (NIC) exhibiting a higher (in terms of more negative values) contribution from electrostatic energy (GBSA eel) in both subtypes of nAChR (Table 3). Conversely, neonicotinoid parent compounds (IMI and THIAC) on average showed less negative binding energy values. A potential explanation for this observation probably lies in the structure of both protein and ligands, because the protonated nitrogen subgroup can be stabilized via cation–π interactions with aromatic residues that are nestling the nicotinoid compounds. Also, the endogenous ligand, acetylcholine, possesses such a moiety, which is an additional rationale for the observed effect. Furthermore, when entropy calculations using a quasi-harmonic approximation [43] were incorporated into the estimation of the binding energy (total delta S entropy), nicotinoids reported less negative values in this term (Table 3). This indicates an additional positive entropic effect contributing to an increased binding affinity.

#### 2.5.3. Quantification of Uncertainty for Binding Energy Predictions

In order to assess the uncertainty of the predicted binding energy approximates for NIC, DCNT, DCNTO, DNIMI, IMI, and THIAC for both studied nAChR subtypes α7 and α3β4, we calculated the mean value of MM-GBSA dG, the 95% confidence interval, and the standard uncertainty (s(x)/√n). This statistical analysis of the so-called observable, MM-GBSA dG, which was derived from four independent simulations (*n* = 4), aims to express the uncertainty of the calculations (Table 4). This analysis is amongst the best practices for the quantification of uncertainty of molecular simulations [44]. Furthermore, it facilitates interpretation and is especially needed in the context of toxicological risk assessments, where regulatory authorities need to make a final decision for each compound. Therefore, Table 4 provides an overview of the statistical uncertainty of the derived MM-GBSA delta G binding energies. The predictions of the desyano-metabolites of THIAC, DCNT, and DCNTO have the lowest standard uncertainty among the nicotinoid metabolites, as their binding energies are in the narrowest range. This strengthens the reliability of their prediction for increased binding affinity with human nAChR compared to the parent compound, which therefore represents a similar pattern to IMI and DNIMI that has been observed in a previous study [17]. Moreover, all nicotinoid metabolites, DCNT, DCNTO, and DNIMI, report even more negative mean MM-GBSA dG binding energies when compared to the reference compound nicotine. This indicates that nicotinoid derivatives are predicted to be more active than nicotine itself with respect to nAChR binding, which is contrary to the predictions for the neonicotinoid parent compounds.

### 2.6. Single Cell Calcium Measurements

LUHMES cells are a suitable in vitro system for the quantification of neonicotinoid signaling through nAChRs. In our previous study, we showed that parent neonicotinoids are less potent than nicotine (NIC) itself by a factor of ten [18]. Docking studies, molecular dynamics simulations, and binding energy calculations propose that the metabolite descyano-thiacloprid (DCNT) shows a potency similar to NIC. Therefore, we investigated if DCNT exhibits a similar response in our Ca^2+^ imaging assay to NIC or the parent compound thiacloprid (THIAC). In Figure 4, the results are depicted as % reactive cells ± SEM. NIC, at its highest tested concentration of 100 µM, evoked a response from approximately ~42% of reactive cells. NIC caused an increase in reactive cells starting from 0.1 µM, with approximately ~9% reactive cells in comparison to the untreated control (differentiation medium) with approximately ~1% reactive cells. In comparison, the neonicotinoid THIAC, at its highest tested concentration of 100 µM, evoked a response from approximately ~15% of reactive cells. The parent compound caused an increase in reactive cells at 1 µM with approximately ~2% reactive cells compared to the untreated control. The descyano-metabolite DCNT, at its highest tested concentration of 100 µM, evoked a response from approximately ~40% of reactive cells. DCNT caused an increase in reactive cells starting from 0.1 µM, with approximately ~7% reactive cells compared to the untreated control (Figure 4). This leads to the conclusion that DCNT shows a comparable potency and concentration response to nicotine and is more potent than its parent compound THIAC.

## 3. Materials and Methods

### 3.1. Molecular Docking Studies

The crystal structures of nAChR isoforms α7 (PDB-ID: 7kox, 7koq, 7koo [11]) and α3β4 (PDB-ID: 6pv7, 6pv8 [37]) were downloaded from the Protein Data Bank [45]. For the protein structures of nAChR isoforms α7 and α3β4, which contain a co-crystallized water molecule in the binding site, two versions were generated, and the presence or absence of a water molecule is indicated as ws1 and ws2, respectively. This results in eight protein structures overall, namely 7kox-ws1, 7kox-ws2, 7koq, 7koo, 6pv7-ws1, 6pv7-ws2, 6pv8-ws1, and 6pv8-ws2 [11,37]. For each complex, only two essential chains, A and B, that form the binding site at their extracellular interface, were retained. Both proteins and ligands (Appendix A) were prepared at pH 7.4 ± 0.5 using Protein Preparation Wizard and Ligprep [46,47], keeping the remaining settings at their default values. 

For the complexes co-crystallized with a small ligand (PDB ID: 7kox, 7koq [11] and 6pv8, 6pv7 [37]), the grid box center was defined by its respective coordinates. For the resting state conformation of nAChR alpha7 (PDB-ID: 7koo [11]) that binds the alpha-bungarotoxin peptide in the orthosteric site, the center of the binding site was defined by the following residues: Tyr92, Asn93, Ser147, Trp148, Ser149, Tyr150, Tyr187, Glu188, Cys189, Cys190, Lys191, Tyr194, Pro195 and Ile53, Trp54, Asn106, Val107, Leu108, Gln116, Tyr117, and Leu118 from the principal (chain A) and complementary subunit (chain B), respectively. The docking protocol uses the Glide XP (extra precision) scoring function [26] within the virtual screening workflow of Maestro (Schrödinger Release 2021-1: Maestro, Schrödinger, LLC, New York, NY, USA).

The suitability of the docking algorithm was assessed through a redocking procedure of the co-crystallized ligand and validated through the RMSD calculation. If the docking pose lay within 2 Angstroms (Å) of the crystallized pose, the docking protocol was considered as validated. This procedure was followed by an ensemble docking approach, where multiple conformations of the protein were used [48].

The representative final docking modes in each protein structure were selected from the top 5 ranked poses based on the docking score and a visual inspection. The resulting docking poses were visualized with Maestro (Schrödinger, LLC, New York, NY, USA) and Pymol (version 2.5) [30].

Considering the fact that docking studies have the tendency to under-sample the conformational space of ligands in a target binding site, we aimed to enhance conformational sampling via molecular dynamics simulations. To enhance the scoring accuracy, the binding energies of both docked and MD-simulated structures were calculated.

### 3.2. Binding Free Energy Calculations of Docking Results

Post-docking MM-GBSA (molecular mechanics-generalized Born Surface Area model) minimization was performed using Prime-MMGBSA [41]. The latest OPLS4 force field [40] was applied to 6 Å steps of the selected binding poses using the VSGB solvation model [49] to yield insights concerning different energetic contributions to ligand binding.

### 3.3. Molecular Dynamics (MD) Simulation

The representative complexes obtained from the ensemble docking method of IMI, THIAC, DNIMI, DCNT, DCNTO, and nicotine in the nAChR isoforms α3β4 and α7 were evaluated through MD simulations. Since the orthosteric binding site of interest is located approximately 20 Å away from the membrane domain, the protein was cut just before the TMD, so that only the ECD part was used for the simulation. The MD engine of choice was NAMD software version 2.14 (University of Illinois at Urbana-Champaign, Urbana, IL, USA), [50], freely available for academic use.

Preparation and parametrization of the protein–ligand complexes were performed by applying the Amber force field (ff14SB for protein description [51]), which was implemented in the AmberTools22 package [52] [53]. Contrary to the neonicotinoids, nicotinoid metabolites were assumed to be protonated at physiological pH [29]; hence, they were assigned a positive net charge using the am1-bcc method [54] of the antechamber module.

GAFF (generalized Amber force field [55]) was used to assign atom types, and the complexes were solvated in a cubic TIP3P [56] water box of 10 Å. An appropriate number of sodium and chloride atoms was added to achieve a neutralized system. After 10,000 steps of initial system minimization, the system was heated up to 300 K with constant volume for 250 ps by applying the Langevin method and thermostat [57]. Particle mesh Ewald (PME) was used for electrostatics [58,59], Periodic Boundary Conditions (PBC) were applied to approximate an infinite system, and the SHAKE algorithm [60] was used for hydrogen bond constraints. This was followed by a 250 ps equilibration run for achieving a 1atm target pressure using Nosé–Hoover Langevin piston method control [61,62]. Finally, each complex was simulated for 50 ns, using a 2 fs integration timestep, resulting in a total simulation time of over 1 µs (6 ligands × 2 isoforms × 2 starting conformations × 50 ns). The resulting trajectory from the production run contains 500 frames, where 10 frames are equivalent to 1 ns simulation time because an integration time step of 2 fs was used, and the frequency for saving the trajectory was every 10,000 steps. Van der Waals (vdW) interactions were treated using the Lennard-Jones potential, which applies an analytical tail correction to the reported vdW energy and virial equal to the amount lost due to switching and cut-off of the LJ potential. vdW interactions were truncated at the cut-off distance, set to 9.0. A cut-off introduces a discontinuity in the potential energy at the cut-off value. As forces are computed by differentiating potential, a sharp difference in potentials may result in nearly infinite forces at the cut-off distance. For visual inspections and analysis of hydrogen bonds (the cut-off for donor–acceptor distance was 3.5 Å and 30° for the donor–hydrogen–acceptor angle), VMD [63] was used. Additional RMSD (root mean square deviation) analysis was performed using a script obtained from the MD Analysis toolkit [64] that also enabled pairwise RMSD calculations. For the 1D RMSD plots, the RMSD of the molecules was calculated using the built-in analysis tool “RMSD Trajectory Tool” from VMD. The analyzed data were extracted and saved for better visualization. The plots were generated using the python libraries matplotlib and pandas [65,66].

#### Postprocessing Analysis of MD Simulations

The MM-GBSA approach [67] was also applied as a post-processing method to the MD simulation using cpptraj available in AmberTools [50,68]. Again, an implicit solvent model was used. Intermediate snapshots were extracted from the MD trajectory to account for conformational changes that occurred during the simulations [35]. Additionally, a potential entropic effect was computed by a quasi-harmonic approximation [43].

### 3.4. In Vitro Approach

#### 3.4.1. LUHMES Cell Culture

LUHMES cells were cultured as described before by Scholz et al. [69]. The 96-well plates (Sarstedt, Nümbrecht, Germany) were coated with 100 µL PLO (50 µg/mL) (Sigma-Aldrich, Merck, Darmstadt, Germany) in PBS (*w*/*o* Ca^2+^ and Mg^2+^) into each well. After one night at 37 °C, the solution was discarded. The multi-well plates were then washed three times with PBS. Afterwards, 100 µL of fibronectin (1 µg/mL) (Sigma-Aldrich, Merck, Darmstadt, Germany) and laminin (1 µg/mL) (Sigma-Aldrich, Merck, Darmstadt, Germany) in PBS were added to each well and incubated overnight at 37 °C. The solution was discarded directly before the use of the multi-well plates.

LUHMES cells were maintained in T75 flasks (Sarstedt, Nümbrecht, Germany) in a proliferation medium. The medium consisted of advanced DMEM/F12 medium supplemented with N2 supplement (1×) (Invitrogen, Karlsruhe, Germany), glutamine (2 mM) (Gibco, Rockville, MD, USA), and recombinant basic fibroblast growth factor (40 ng/mL) (bFGF, R&D Systems, Minneapolis, MN, USA). Cells were split every two days when reaching 80% confluency. Before splitting, the cells were washed once with DPBS, then detached with 0.05% trypsin (Sigma-Aldrich, Merck, Darmstadt, Germany), collected in non-supplemented medium centrifuged at 340× *g* for 4 min, resuspended in the medium, and counted with a Neubauer chamber. The cells were then seeded in a T75 flask with 15 mL of proliferation medium or differentiation medium, respectively (day −1, pre-differentiation). Three million cells were used for maintenance, three million were used for pre-differentiation, and nine million were used for differentiation.

One day after seeding (day 0), the proliferation medium was exchanged for the differentiation medium. The differentiation medium consisted of advanced DMEM/F12 medium supplemented with glutamine (2 mM) (Gibco, Rockville, MD, USA), cAMP (1 mM) (Sigma-Aldrich, Merck, Darmstadt, Germany), tetracycline (2.25 µM) (Sigma-Aldrich, Merck, Darmstadt, Germany), and glial cell-derived neurotrophic factor (2 ng/mL) (GDNF, Bio-Techne, Minneapolis, MN, USA). The proliferation medium was aspirated, the flask was washed with 10 mL DPBS, and afterward 15 mL of differentiation medium was added.

LUHMES cells were then cultivated for two days in the T75 flask and then seeded into 96-well plates at a density of 60,000 cells/well (206,897 cells/cm^2^) for experiments on day 9. On days 4, 6, and 8, half a medium exchange was performed with fresh differentiation medium. Calcium signaling experiments were performed on day 9.

#### 3.4.2. LUHMES Ca^2+^ Imaging

Ca^2+^ imaging was performed using a Cellomics Arrayscan VTI HCS Reader (Thermo Fisher Scientific, Waltham, MA, USA) equipped with an automated pipettor and an incubation chamber. Measurements were recorded at nominal 37 °C and 5% CO_2_. The Cellomics Arrayscan allows for the recording of indirect changes in [Ca^2+^]_i_ via a Ca^2+^-sensitive fluorescent dye. The Cellomics Arrayscan VTI HCS Reader acquires the fluorescence signal of one 96-well plate at a time. The integrated pipet unit allows a controlled compound administration into one well. Cells were imaged as fast as possible for 45 s at approximately 2 fps. Compounds were administered automatically after 10 s of baseline recording. The images were exported as 16-bit .tiff image files and analyzed in CaFFEE software version 2 [70].

LUHMES cells were differentiated for 2 days in a T75 flask. On differentiation day 2, cells were seeded into 96-well plates at a density of 60,000 cells/well (206,897 cells/cm^2^) and cultivated until differentiation day 9, if not indicated otherwise. Cells were incubated with Cal-520 AM (AAT Bioquest) for 1 h at a concentration of 5 µM at 37 °C. The fluorescence Ca^2+^ indicator solution contained 1 µM/mL HOECHST-33342 (H-33342) (Sigma-Aldrich, Merck, Darmstadt, Germany); for nuclear detection, PNU-120596 (10 µM) (Sigma-Aldrich, Merck, Darmstadt, Germany); an allosteric modulator for the α7 nicotinic receptor, Probenecid (1.54 mg/mL) (Thermo Fisher Scientific, Karlsruhe, Germany); an inhibitor of organic anion transporters located in the cell membrane; and Pluronic^TM^-F127 (0.4%) (Thermo Fisher Scientific, Karlsruhe, Germany), a non-ionic tensid-polyol which helps with the dispersion of the dye.

#### 3.4.3. Calcium Fluorescent Flash Evaluating Engine (CaFFEE)

The CaFFEE enables the user to process and evaluate large quantities of image data. It automatically identifies individual cells in mixed, heterogeneous populations and evaluates their fluorescent signal. It enables the evaluation of the influence of a treatment on the [Ca^2+^]_i_ of hundreds of cells. The data can be exported in spreadsheet format. Moreover, the image data can be processed for an optimized visual representation of the time-lapsed image data, which can be explored by setting the parameters for semi-automated data processing [70]. The CaFFEE identifies cells automatically by additional staining with HOECHST-33342 for structural features. After identification of a cell, the CaFFEE defines each cell as a region of interest. The average fluorescent intensity of these pixels is measured over a series of pictures, thereby obtaining time-dependent fluorescent values for each cell. This information is then converted into curves from which different parameters can be obtained.

With the CaFFEE, the timepoint of peak fluorescence can be obtained. Fluorescent data for baseline recording allowed for the automated assessment of the ground state (F_0_) and peak timepoint (F_1_) for all cells. The difference between base level and peak level in fluorescence, ΔF = F_1_ − F_0_, was used for further data analysis.

The noise-level-based threshold, mean (ΔF) + 3 × SD (ΔF), defining a reactive or non-reactive cell, was determined as follows: The mean ΔF value was calculated from all wells that received a negative stimulus (differentiation medium). A cell was defined as reactive when ΔF_stimulus_ > threshold or non-reactive when ΔF_stimulus_ < threshold.

## 4. Conclusions

Within this study, a common hypothesis was applied stating that noncovalent interactions of protein–ligand complexes can be used to characterize the binding mode and to explain differences in binding affinities. In this context, a higher affinity is due to more beneficial interactions being formed in the binding site.

When an electronegative moiety is lost through metabolism (DNIMI, DCNT, DCNTO), either via environmental causes or enzymatically [14], an increased binding affinity towards human nAChRs can be expected, which was also predicted by the structure-based approaches used in this study. Other metabolic changes, such as reduction to an additional double bond, e.g., imidacloprid-olefin, would not cause equivalent drastic changes in affinity, which has also been shown in our previous study [17]. The underlying mechanism may be based on the increased stability of the nicotinoids in the binding site, which is also reflected in binding energy estimates (including an entropic effect) and also in more favorable interactions. Cation–π interactions seem to be one of the main driving factors for the increased stability of nicotinoid compounds in a binding site [34]. 

For a description of the binding mode of neonicotinoid parent compounds, it is important to consider that more than one binding mode is likely to exist in human nAChRs. This is against the common hypothesis of only one representative binding mode, which is often reported from co-crystallization experiments. Therefore, the issue arises that during crystallization, multiple factors like detergents and freezing techniques influence the derived structural models, which also have to be considered when, e.g., knowledge-based approaches for binding mode reconstructions are applied [71]. This reinforces the need for additional methods for binding mode elucidations, such as molecular docking and MD simulations, since they also, at least partly, account for physiological protein flexibility. When analyzing the results of structure-based computational approaches, it is also crucial to consider that different scoring functions have an influence on ranking correlations and the selection of “representative” poses. Nevertheless, these methodologies present a benefit in that they generate 3D descriptors, since the triggering action of protein binding is also three-dimensional in nature. Therefore, contacts with Trp148, Tyr92 and Trp149 residues and the Tyr93 residue in α7 and α3β4, respectively, could serve as advanced structural alerts or as descriptive inputs for machine learning models to indicate increased binding affinity to nAChRs that are involved in potential AOPs (adverse outcome pathways) causing DNT (developmental neurotoxicity). As an extension to this, transferring the binding modes to toxophores and further applying them as advanced structural alerts for the classification of moderate/acute effects on human nAChRs could be an additional application of the work presented in this study. Furthermore, these structural insights help in understanding one of the MIEs (molecular initiating events) for DNT. Importantly, different mammalian nAChR subtypes exist and in silico models can present insights into compound–receptor interactions for different subunits, as presented here for α3β4 and α7. However, in order to gain confirmation that the docked compounds contact the discussed residues, photoaffinity labelling experiments could provide the final proof. Alternatively, a radioligand binding assay could also confirm the binding of DCNT, DNCT, and THIAC to nAChR α3β4. 

Using the available knowledge, we constructed here a putative AOP that links nAChR activation (by, e.g., neonicotinoids) to DNT (Figure 5). While the initial steps/events are clearly defined, later KEs in this preliminary AOP have large uncertainties. Why do we nevertheless feel that such an AOP is justified? The main reason is that there is solid evidence that exposure to nicotine does indeed lead to adverse outcomes related to classical DNT or to neural crest defects. This means that the beginning and end of such an AOP are documented. Nicotine-induced neural crest disturbances are exemplified by craniofacial malformations [72]. Examples of DNT effects are structural alterations in brain regions (especially catecholamine systems), cognitive deficits, cortical dysfunctions, and an increased incidence of attention deficit/hyperactivity disorder (ADHD) [73,74,75,76,77,78]. In addition to this toxicological evidence for the existence of an AOP with nAChR linked to DNT, there is also technical evidence that such types of AOP may be constructed and accepted. Other AOPs for DNT, with receptor interactions as the MIE, exist, e.g., for NMDA (N-methyl-D-aspartate) receptors [23,24]. The adverse outcomes, as observable in the human population or in animal studies, are not a highly defined disease symptom or a quantifiable pathology. They are better described as a large spectrum of potential cognitive performance deficits, which vary considerably between species and individuals.

The concept of endophenotypes as one form of adverse outcome helps to better define AOPs related to DNT. This concept is very broadly used in psychiatry and has been introduced into toxicology to allow the establishment of more robust AOPs [79,80]. The basic assumption is that any external symptom (often hard to quantify) must be linked to an altered network connectivity (structural or functional) in the brain [81]. This endophenotype may be assessed by classical toxicological methods (histology or neurophysiology). This would be more difficult for, e.g., attention deficits or an altered executive function. One endophenotype may manifest in several exophenotypes and thus explain the broad spectrum of potential adverse events. For instance, dopamine network disturbance in basal ganglia may lead to hyperlocomotion and restlessness in rodents, but to delusions in humans. For practical purposes, our suggested AOP has two sequential AOs. This is not uncommon, as it is also the case, e.g., in AOP 3 (Inhibition of the mitochondrial complex I of nigrostriatal neurons leads to parkinsonian motor deficits [82]). The two AOs (endophenotype and exophenotype) in this AOP are a loss of nigrostriatal dopaminergic neurons (endophenotype) and parkinsonian motor deficits (exophenotype).

With these considerations, and since our experimental validation shows that also a key event (KE) downstream of the MIE is activated by the metabolite DCNT, we provide an alert for the potential involvement of thiacloprid in DNT. For further validation, investigations of KE3, in which we postulate key neurodevelopmental processes (KNDP) are altered, will be important. A promising tool will be assays using microelectrode arrays (MEAs) to examine network functionality, which is closely linked to an altered endophenotype.

Herein, methodologies were applied that are at the intersection of drug design and toxicology and that have been carefully evaluated. We paved the way for the integration of structure-based tools with validation by in vitro methods as NAMs in NGRAs. A further goal could be the specific tailoring of an in vitro and in silico test battery suitable for a specific research direction. In this context, regulatory acceptance could be increased by quantifying the uncertainty of an NAM approach and by eventually being able to perform better human risk assessments.

## Figures and Tables

**Figure 1 ijms-24-13170-f001:**
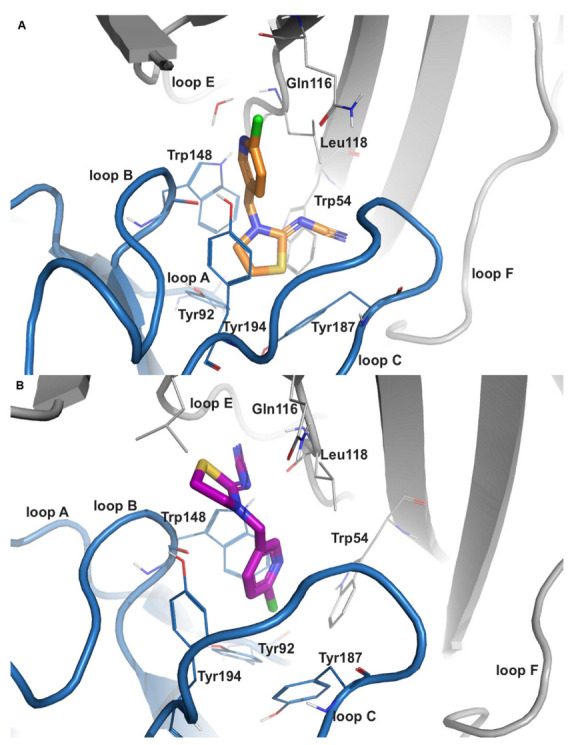
Representative binding modes of thiacloprid (THIAC) in human nAChR α7. (**A**) The common binding mode of THIAC (orange carbon atoms) in the nAChR α7 structure (PDB-ID: 7kox) is similar to a homologous co-crystallized structure in AChBP; the nitrogen of the chloro-pyridine ring faces towards loop E (and the central pore of the receptor), whereas the electronegative cyano-group is pointing towards the tip of loop C. (**B**) Inverted binding mode of THIAC (purple carbon atoms) in the nAChR α7 structure (PDB-ID: 7koq), where the cyano-group is oriented towards loop E and the chloro-pyridine ring is directed towards the transmembrane domain. The protein backbone of the receptor is shown in a ribbon representation with the principal and complementary subunits in blue and gray, respectively.

**Figure 2 ijms-24-13170-f002:**
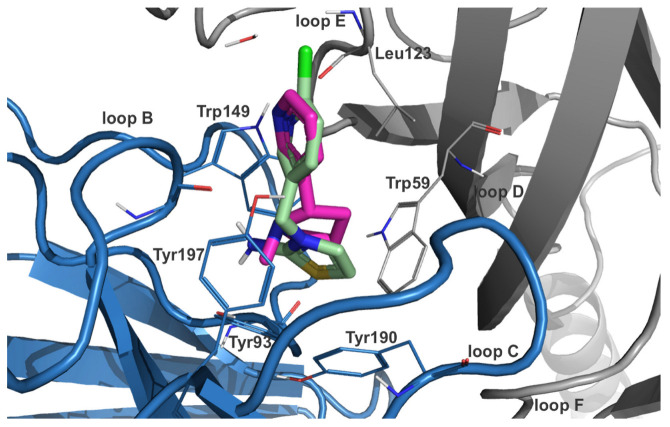
Docked pose of DCNTO (pale green carbon atoms) superimposed with co-crystallized nicotine (pink carbon atoms) in nAChR isoform α3β4 (PDB-ID: 6pv7). The position of the chloro-pyridine ring is in alignment with the heteroarylic ring of nicotine and the positively ionizable nitrogen substructure of the thiazol ring is stabilized via favorable interactions with residues from the aromatic box (Tyr93, Trp149, Tyr190, Tyr197, and Trp59 from the principal (blue ribbon) and complementary (gray ribbon) subunit, respectively).

**Figure 3 ijms-24-13170-f003:**
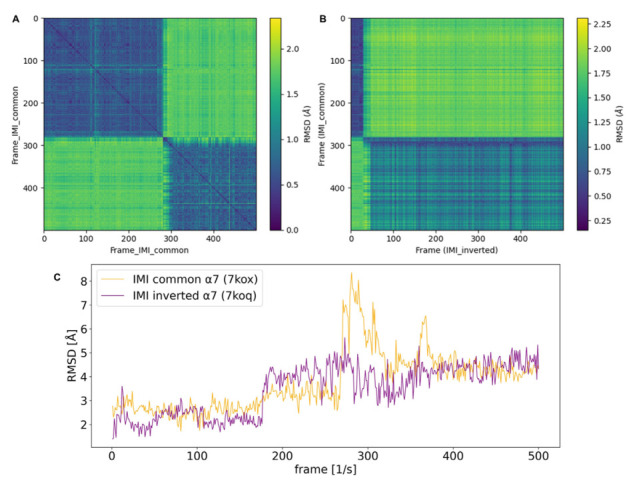
RMSD plots of imidacloprid (IMI) in structures of human nAChR α7 (7kox, 7koq). (**A**) Plot of the pairwise RMSD of the 50ns MD trajectory (500 frames) of the common binding mode of IMI in nAChR α7 (PDB-ID:7kox) against itself. After 280 frames, which is equivalent to 28ns simulation time, the only major conformational transition occurs. (**B**) Plot of the pairwise RMSD of the trajectories of the common binding mode of IMI (IMI_common; 7kox) which was compared to the trajectory of IMI in an inverted binding mode (IMI_inverted; 7koq). IMI_common converges towards the conformational space of IMI_inverted during the last 310 simulation frames. (**C**) One-dimensional RMSD plot of IMI equivalent to (**A**); a major conformational transition of the common binding mode of IMI (orange curve) occurs after 280 frames. The inverted binding mode of IMI (purple curve) shows one transition occurring at 180 simulation frames and reports an overall lower mean RMSD than the common binding mode (see also Table 2).

**Figure 4 ijms-24-13170-f004:**
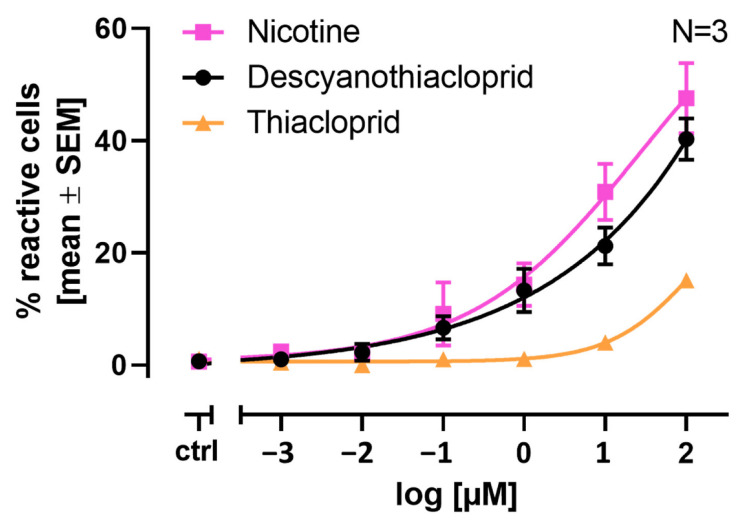
Concentration response comparison between nicotine (NIC), thiacloprid (THIAC), and descyano-thiacloprid (DCNT). LUHMES cells were seeded on differentiation day 2 into 96-well plates at a density of 60,000 cells/well and differentiated until DoD9. A total of 75 µL of the differentiation medium was exchanged with 25 µL of Cal520-AM (5 µM) Ca^2+^ indicator solution 1h before measurement. The number of cells responding to the respective stimulus is shown in the mean percentage of reactive cells ± SEM. Cells were recorded as reactive when their response exceeded the mean of the control plus three times the standard deviation of the untreated control. Concentration–response curves for the treatment with NIC, THIAC, and DCNT at concentrations ranging from 0.001 µM to 100 µM. The curves shown are NIC in pink squares, THIAC in orange triangles, DCNT in black circles. Number of biological replicates N = 3.

**Figure 5 ijms-24-13170-f005:**
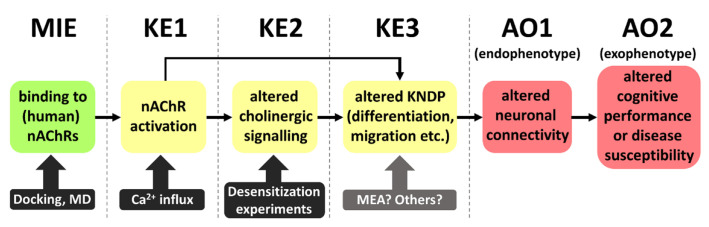
Putative AOP (adverse outcome pathway) linking nAChR binding to potential adverse outcomes related to DNT. The AOP is based on the assumed mode of action of nicotine, which might apply also to certain neonicotinoids and their metabolites. KE3 might be triggered via KE1 either directly or via KE2. At present, KE3 and the adjacent key event relationships are the most uncertain elements of the putative AOP. More information on these processes is necessary. MD: molecular dynamics; MIE: molecular initiating event; KE: key event; AO: adverse outcome; MEA: microelectrode array.

**Table 1 ijms-24-13170-t001:** Spearman ranking correlation from ensemble docking analysis of docking scores and dG binding energy of 15 (neo-)nicotinoid compounds (Appendix A) and experimentally derived IC_50_ values in nAChR α7 (7kox, 7koq, 7koo) and α3β4 structures (6pv7, 6pv8).

	α7				α3β4			
PDB-ID-water set	7kox-ws1	7kox-ws2	7koq	7koo	6pv7-ws1	6pv7-ws2	6pv8-ws1	6pv8-ws2
Docking score	0.75	0.7	0.64	0.0089	0.75	0.66	0.81	0.74
dG bind	0.57	0.54	0.6	0.3	0.53	0.38	0.43	0.45

**Table 2 ijms-24-13170-t002:** RMSD calculations from MD analysis. Protein structures of the starting conformation of the ligands are indicated with PDB-ID. If a water molecule was present in the initial protein–ligand complex obtained from docking studies, it is indicated as PDB-ID-ws1, whereas PDB-ID-ws2 indicates that the co-crystallized water molecule was removed. PDB-ID alone indicates that there was no co-crystallized water molecule.

	**Compound**	**PDB-ID**	**Mean RMSD**	**Average Mean RMSD**
α7	IMI	7kox-ws2	3.78	2.96
		7koq	2.14	
	DNIMI	7kox-ws1	2.34	2.03
		7koq	1.73	
	THIAC	7kox-ws1	3.87	3.15
		7koq	2.43	
	DCNT	7kox-ws1	1.91	1.60
		7koq	1.29	
	DCNTO	7kox-ws1	1.73	1.78
		7koq	1.83	
	NIC	7kox-ws2	1.86	1.82
		7koq	1.78	
α3β4	IMI	6pv7-ws2	3.47	4.15
		6pv8-ws2	4.82	
	DNIMI	6pv7-ws2	1.87	2.17
		6pv8-ws1	2.46	
	THIAC	6pv7-ws1	2.53	2.78
		6pv8-ws2	3.03	
	DCNT	6pv7-ws2	1.73	1.82
		6pv8-ws1	1.91	
	DCNTO	6pv7-ws2	1.37	1.38
		6pv8-ws1	1.38	
	NIC	6pv7-ws2	1.43	1.61
		6pv8-ws2	1.78	

**Table 3 ijms-24-13170-t003:** MM-GBSA binding energy calculations of the MD trajectories, executed by MMPBSA.py. Protein structures of the starting conformation of the ligands are indicated with PDB-ID, whether a water molecule was present in the initial structure or not is indicated as ws1 or ws2, respectively.

	**Ligand**	**PDB-ID**	**dG Binding Energy**	**vdW Energy**	**Eel Energy**	**dG + Entropy**	**Total Delta S Entropy**
α7	IMI	7kox-ws2	−27.08	−35.13	−10.15	8.43	−35.51
		7koq	−36.44	−42.91	−10.04	−1.13	−35.31
	DNIMI	7kox-ws1	−36.31	−35.66	−163.17	−3.34	−32.96
		7koq	−36.31	−36.10	−169.96	−3.61	−32.70
	THIAC	7kox-ws1	−33.77	−42.55	−1.12	−0.83	−32.94
		7koq	−33.50	−39.74	−12.52	0.12	−33.63
	DCNT	7kox-ws1	−39.72	−39.44	−163.88	−7.15	−32.57
		7koq	−37.62	−37.05	−156.58	−4.81	−32.81
	DCNTO	7kox-ws1	−35.33	−35.57	−155.57	−4.77	−30.56
		7koq	−36.77	−36.91	−153.18	−5.14	−31.63
	NIC	7kox-ws2	−33.72	−31.41	−143.67	−3.29	−30.42
		7koq	−35.79	−31.13	−165.53	−6.32	−29.47
α3β4	IMI	6pv7-ws2	−25.34	−36.44	−11.51	11.86	−37.20
		6pv8-ws2	−24.24	−35.16	−13.67	13.11	−37.35
	DNIMI	6pv7-ws2	−44.21	−38.20	−147.71	−13.18	−31.03
		6pv8-ws1	−37.48	−36.30	−127.50	−4.52	−32.96
	THIAC	6pv7-ws1	−28.50	−34.97	−22.51	4.73	−33.23
		6pv8-ws2	−32.74	−38.60	−22.39	−0.47	−32.27
	DCNT	6pv7-ws2	−35.63	−36.20	−129.06	−3.15	−32.47
		6pv8-ws1	−36.38	−35.38	−124.08	−3.84	−32.54
	DCNTO	6pv7-ws2	−37.23	−37.45	−119.56	−7.83	−29.40
		6pv8-ws1	−37.74	−36.85	−135.41	−6.96	−30.78
	NIC	6pv7-ws2	−36.62	−31.06	−120.78	−6.58	−30.04
		6pv8-ws2	−33.42	−28.69	−120.71	−4.42	−30.00

**Table 4 ijms-24-13170-t004:** Statistical uncertainty of observable (MM-GBSA delta G (dG) binding energy).

**Ligand**	**Mean MM-GBSA dG**	**95% Confidence Interval**		**Standard Uncertainty**
NIC	−34.89	−37.39	−32.38	0.55
DCNT	−37.34	−40.18	−34.49	1.110
DCNTO	−36.77	−38.41	−35.12	1.198
DNIMI	−38.58	−37.80	−35.86	1.374
IMI	−28.27	−37.13	−19.42	1.953
THIAC	−32.13	−35.04	−29.22	1.243

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
