# Peer review of "Structural Insights into Neonicotinoids and N-Unsubstituted Metabolites on Human nAChRs by Molecular Docking, Dynamics Simulations, and Calcium Imaging"

_ijms, 2023, doi:10.3390/ijms241713170_

Round 1

Reviewer 1 Report

Authors investigated the lack of selectivity of neonicotinoid pesticides by running molecular docking, MD simulations and calculation of binding free energy with the human nAChR.  Publication is recommended after minor revisions.

Line 88-90: Remove these lines.

Line 102: Explain how the multiple conformations are obtained.

108: ab34 --> a3b4

110-111: Explain what ws1 is.

Table S1: Add caption to explain ws1, ws2 and PDB ID without these.

Table S1: alpha3 ==> alpha3beta4

Table S1: The commas should be changed to period in the table.

144: ab34 --> a3b4

Table 2: Explain clearly the differences among three conformations. (PDB ID only, PDB ID with ws1, PDB ID with ws2). 

453: Conformation of the protein does not change by simply removing the water. Explain clearly.

Section 3.2 and 3.3: Why the binding free energy was calculated by two force field which are OPLS and Amber? Be consistent.

716~732: Remove it or update it.

Reviewer 2 Report

The present paper discusses the potential toxic effects of neonicotinoid pesticides and their metabolites on human neurons. The study proposes a combination of in silico and in vitro methods to assess the toxic effects of neonicotinoids. First, an ensemble docking study is conducted on nAChR isoforms α7 and α3β4 to identify crucial molecular initiating event interactions. Molecular dynamics simulations and binding energy calculations are then performed to refine the docking poses. While this in silico study is technically correct, it lacks originality and it is prone to errors and inconsistencies. Nevertheless, given the complexity of the system and the targeted problem, most methods would encounter similar challenges. Although more accurate and less "automatic" docking methods could be employed, the current study remains valid, and the procedure is appropriate.

The writing is clear and well-organized. The presentation of the results is affective and easy to follow.

Reviewer 3 Report

The manuscript “Structural insights of neonicotinoids and N-unsubstituted metabolites on human nAChRs by molecular docking, dynamics simulations, and calcium imaging” by Karin Grillberger et al. combines in silico and in vitro methods to investigate the interaction of neonicotinoid pesticides with human nicotinic acetylcholine receptors (nAChRs). The study confirms the agonistic effect of a specific metabolite called descyano-thiacloprid on human neurons and proposes a new approach for the next-generation risk assessment that may reduce the reliance on animal testing. The computational studies are robust, and the authors have considered the conformational entropy upon ligand binding in their MM-GBSA calculations, which is often ignored in docking studies. Also, the presentation is very good. I think this well-organized manuscript is worthy of publication in Int. J. Mol. Sci. I only have some minor concerns for the authors to address before I can endorse its publication.

1. The Materials and Methods section needs some clarifications and additional references.

1) Cite [W. L. Jorgensen, et al. J. Chem. Phys. 198379 (2), 926–935] for the Jorgensen’s original TIP3P water model.

2) Cite [A. T. Brünger. X-PLOR, Version 3.1, A System for X-ray Crystallography and NMR; Yale University Press: New Haven and London, 1993] for the Langevin thermostat.

3) Cite [G. J. Martyna, et al. J. Chem. Phys. 1994101 (5), 4177–4189; S. E. Feller, et al. J. Chem. Phys. 1995103 (11), 4613–4621] for the Nosé–Hoover Langevin piston barostat.

4) Cite [T. Darden, et al. J. Chem. Phys. 199398 (12), 10089–10092; U. Essmann, et al. J. Chem. Phys. 1995103 (19), 8577–8593] for PME.

5) Cite [J.-P. Ryckaert, et al. J. Comput. Phys. 197723 (3), 327–341] for the SHAKE algorithm.

6) How did the authors treat the van der Waals (vdW) interactions? Did the authors potential- or force-switch the vdW? What was the switch range?

7) The statement “hydrogen bonds (cut-offs for distance and angle are 3.5 AÌŠ and 30Ëš” is not clear enough. Did the 3.5 Å cutoff pertain to the donor–hydrogen or donor–acceptor distance? Which angle was the 30Ëš cutoff applied to?
